# The Kelch Repeat Protein VdKeR1 Is Essential for Development, Ergosterol Metabolism, and Virulence in *Verticillium dahliae*

**DOI:** 10.3390/jof10090643

**Published:** 2024-09-09

**Authors:** Wen-Li Xia, Zhe Zheng, Feng-Mao Chen

**Affiliations:** Co-Innovation Center for Sustainable Forestry in Southern China, College of Forestry and Grassland, College of Soil and Water Conservation, Nanjing Forestry University, Nanjing 210037, China; xiaxia@njfu.edu.cn (W.-L.X.); zhengzhe@njfu.edu.cn (Z.Z.)

**Keywords:** *Verticillium* wilt, kelch-domain protein, fungal development, pathogenicity, ergosterol synthesis pathway

## Abstract

*Verticillium dahliae* is a soil-borne fungal pathogen that can cause severe vascular wilt in many plant species. Kelch repeat proteins are essential for fungal growth, resistance, and virulence. However, the function of the Kelch repeat protein family in *V. dahliae* is unclear. In this study, a Kelch repeat domain-containing protein DK185_4252 (VdLs.17 VDAG_08647) included in the conserved *VdPKS9* gene cluster was identified and named VdKeR1. Phylogenetic analysis demonstrated a high degree of evolutionary conservation of VdKeR1 and its homologs among fungi. The experimental results showed that the absence of *VdKeR1* impaired vegetative growth, microsclerotia development, and pathogenicity of *V. dahliae*. Osmotic and cell wall stress analyses suggested that *VdKeR1*-deleted mutants were more tolerant to NaCl, sorbitol, CR, and CFW, while more sensitive to H_2_O_2_ and SDS. In addition, analyses of the relative expression level of *sqe* and the content of squalene and ergosterol showed that *VdKeR1* mediates the synthesis of squalene and ergosterol by positively regulating the activity of squalene epoxidase. In conclusion, these results indicated that *VdKeR1* was involved in the growth, stress resistance, pathogenicity, and ergosterol metabolism of *V. dahliae*. Investigating *VdKeR1* provided theoretical and experimental foundations for subsequent control of *Verticillium* wilt.

## 1. Introduction

Ergosterol, one of the major components of fungal membranes, is involved in numerous biological functions [1]. Extensive literature reports the fundamental contribution of sterols to fluidity, permeability, microdomain formation, protein functionality, and membrane activities [2,3,4,5]. These reports indicate that ergosterol and its biosynthetic pathway are essential for fungal growth. In the ergosterol pathway, the biosynthesis of ergosterol starts from acetyl-CoA and passes through critical intermediates such as squalene, 2,3-epoxysqualene, and lanosterol [6]. Steps prior to squalene formation are important for pathway regulation, and early intermediates are metabolized to produce other essential cellular components [7]. The genes mediating steps from lanosterol to ergosterol are essential in *Saccharomyces cerevisiae*. Their deletion causes ergosterol deficiency and the accumulation of a series of ergosterol precursor compounds [8]. Ergosterol deficiency leads to abnormal function of fungal cell membranes, cell rupture, and eventual cell death [9].

Squalene epoxidase (SQE) is essential in the biosynthetic pathway and belongs to the flavoprotein monooxygenase family, which is involved in various oxidation reactions [10,11]. SQE oxidizes squalene to squalene epoxide [1], which participates in the final biosynthesis of cholesterol and ergosterol [12], and is the target of antifungal drugs such as terbinafine [7]. Terbinafine has a specific inhibitory effect on SQE and causes treated fungal cells to rapidly accumulate squalene intermediates, which blocks ergosterol biosynthesis and results in fungicidal effects [13]. The generation of high concentrations of squalene inside treated cells is closely associated with the fungicidal action, as squalene is believed to disrupt fungal membrane function and cell wall synthesis [14].

The biosynthesis of ergosterol is closely associated with fungal growth and pathogenicity. When treated with the SQE-specific inhibitor terbinafine, the SQE activity and ergosterol content of *Candida albicans* are reduced, resulting in significant inhibition of hyphal growth and a decrease in pathogenicity [13]. Similarly, antifungal drugs that target the key enzyme Erg11A can block the ergosterol synthesis pathway of *Aspergillus fumigatus*, resulting in slower fungal growth and reduced pathogenicity [8]. The SQE activity in the *Sporisorium scitamineum SsCI80130* knockout mutant is significantly reduced compared to the WT and complemented strains. Additionally, there is a significant increase in squalene content, a significant decrease in ergosterol content, and a significantly slower mycelium growth rate [15].

Kelch-domain proteins are widespread in eukaryotes [16,17,18]. They contain variable numbers of Kelch repeats (five–seven). Each repeat forms a β-propeller secondary structure. [16,19]. In *Arabidopsis thaliana*, the F-box (KFB) protein containing the Kelch repeat motif FKF1 is involved in photomorphogenesis and flowering time control [17]. The brewing yeast *S. cerevisiae* has two proteins, Gpb1 and Gpb2, each containing seven Kelch repeat motifs. These proteins regulate the activity of protein kinase A (PKA) in response to nutritional requirements [20]. The Tea1 protein in *Ustilago maydis*, which has five Kelch repeat motifs, not only regulates cell morphogenesis but also contributes to the virulence of the corn smut fungus on maize [21]. Studies have shown that Kelch repeat proteins are associated with fungal polarized growth [21,22,23]. In yeast, Ste20p mediates polarisome activation via phosphorylation of the formin Bni1p and also phosphorylates the type I myosin Myo3p for actin patch assembly [24,25]. Sterols also play a crucial role in polarized growth. Ste20p phosphorylates Are2p, the acylCoA sterol acyltransferase responsible for steryl ester [26,27,28,29]. Loss-of-function *ste20* mutants lack filamentous invasive growth and exhibit increased sterol levels [30,31].

We investigated the role of the Kelch repeat protein VdKeR1 in the fungus *V. dahliae*, a soil-borne vascular pathogen that causes *Verticillium* wilt disease. The pathogen can infect over 200 species of dicotyledonous plants [32], including economically important crops such as cotton and tomato and landscape trees such as *Cotinus coggygria* and *Acer truncatum* [33]. The protein family member VdKeR has previously been reported to regulate the virulence of *V. dahliae* towards cotton. Deletion of the *VdKeR* gene in *V. dahliae* results in reduced colony growth rate, a decrease in pathogenicity to cotton seedings, and affects the expression of other pathogenicity-related genes [34]. There have been no reports on the role of this protein family in the regulation of ergosterol biosynthesis. The main objectives of this study were as follows: (1) to identify and localize the VdKeR1 protein in *V. dahliae*, (2) to investigate the role of *VdKeR1* in the growth and development of *V. dahliae*, (3) to examine whether *VdKeR1* was involved in the regulation of the penetration ability and pathogenicity in *V. dahliae*, (4) to explore the regulatory mechanisms of *VdKeR1* on the ergosterol synthesis pathway in *V. dahliae*, and (5) to elucidate the significance of *VdKeR1* in the physiological processes of *V. dahliae*.

## 2. Materials and Methods

### 2.1. Bioinformatics Analysis and Characterization of VdKeR1

DK185-0452 (*VdKeR1*) was selected as a research object from the *VdPKS9* gene cluster based on our previous laboratory research [35]. Kelch repeat proteins information was retrieved from the National Center for Biotechnology Information (NCBI) database. Based on the complete genome data of the At13 strain (unpublished), as well as the results of the comparative genome analysis of the Vd991 strain and the complete genome data of the VdLs.17 strain [36], protein functional domain analysis tools such as InterPro (http://www.ebi.ac.uk/interpro/, accessed on 15 June 2023), Pfam (http://pfam.xfam.org/, accessed on 15 June 2023), and SMART (http://smart.embl-heidelberg.de/, accessed on 15 June 2020) were used. The IBS (Illustrator for Biological Sequences) plotting tool was used to visualize the domain structures of the selected VdKeR1. We used SWISS-MODEL (http://swissmodel.expasy.org/, accessed on 20 July 2023) as a protein structure modelling server and PyMOL (http://www.pymol.org, accessed on 20 July 2023) to illustrate protein structures.

The homologs of *VdKeR1* were identified by BLAST search in the NCBI GenBank database and subsequently downloaded from GenBank. Multiple sequence alignment was performed using Clustal W in Bioedit 7.2 [37]. The phylogenetic tree was then constructed using the maximum likelihood method in MEGA 11.0 [38]. The workflow was as follows: using the WAG+F+G substitution model, pairwise deletion, and bootstrapping 1000 replicates [39]. The phylogenetic tree and alignment were visualized using iTOL (https://itol.embl.de/, accessed on 5 August 2023) and Jalview (https://www.jalview.org/, accessed on 5 August 2023), respectively.

### 2.2. Generation of Gene Deletion Mutants and Complemented Strains

To generate the *VdKeR1* deletion vector, we amplified the upstream and downstream fragments of *VdKeR1* between 800 bp and 1200 bp from the At13 genome DNA. The pCH plasmid (obtained by transformation of pDHt2 [40]) was linearized using the restriction enzymes *EcoR*I (Takara, Nanjing, China) and *Xba*I (Takara). The amplified fragments were then linked to the upstream and downstream regions of the hygromycin resistance gene (*hyg*) by homologous recombination (ClonExpress^®^ II One Step Cloning Kit, Vazyme, Nanjing, China). After introduction of the recombinant plasmid into *Escherichia coli* DH5α (Vazyme) cells, the deletion vector was transformed into *Agrobacterium tumefaciens* strain AGL-1 (Vazyme) using the *Agrobacterium*-mediated transformation (ATMT) method [41,42]. The transformants were selected on PDA plates supplemented with 50 µg/mL hygromycin (Sangon Biotech, Shanghai, China). The pCH deletion vector encoded a hygromycin (Hyg)-resistance gene, so the transformants could grow on PDA medium supplemented with 50 µg/mL hygromycin.

To construct the complemented vector of *VdKeR1*, the coding sequence of *VdKeR1*, including the promoter and terminator regions, was amplified from the At13 genome. It was then cloned into the pCOM vector carrying the G418-resistance gene. The complemented vector was then transformed into the *A. tumefaciens* AGL-1 strain using the ATMT method. Co-cultivation was performed with the Δ*VdKeR1* strains. The transformants were able to grow on PDA medium supplemented with 50 μg/mL geneticin (Sangon Biotech) due to the presence of the G418-resistance gene in the pCOM vector. This confirmed the successful insertion of the *VdKeR1* gene into the transformants, ultimately obtaining complemented strains.

### 2.3. Localization of VdKeR1 by Fluorescence Microscopy Observation

To construct the green fluorescent protein-fused (GFP-fused) strains, the coding sequence (CDS) region of *VdKeR1* without the stop codon and the GFP fragment were amplified and sequenced using high-fidelity enzyme (Vazyme). By homologous recombination (ClonExpress^®^ II One Step Cloning Kit, Vazyme), the amplified fragments were inserted into the Pcom-T0161 vector digested with *Xho*I (Takara). Recombinant plasmids were selected by G418-containing medium. The primers used in this experiment are listed in Appendix A.

Based on previous research protocols [43,44], the recombinant plasmids were introduced into the protoplasts of the wild-type strains, resulting in the generation of GFP-labelled strains. The specific steps were as follows: spores of the wild-type strains were cultured in CM medium for two days, then the mycelium was collected, washed, and digested using an enzymatic method [45]. After 4 h, the digestion products were washed with NaCl buffer and resuspended. The protoplasts were adjusted to a concentration of 5 × 10^6^ conidia/mL using 1 × STC buffer (Vazyme). Next, equal amounts of 2 × STC buffer (Vazyme) and GFP-fused recombinant plasmids were mixed and added to 200 μL of protoplasts. The mixture was allowed to stand at room temperature for 10 min. Following this, 1 mL of 50% polyethylene glycol solution was added and incubated for 20 min. Then 3 mL of TB3 liquid medium (Sangon Biotech) was added and shaken overnight for 10 h to restore the protoplast cell wall. Finally, the recombinant protoplasts were added to TB3 solid medium containing G418 and poured onto Petri dishes. Positive transformants were identified by PCR, and single-spore isolation was performed. The transformants were cultured on PDA and observed for GFP signals using a fluorescence microscope (Zeiss, Jena, Germany).

### 2.4. Evaluation of Colony Morphology, Conidial Production, Melanin Synthesis, and Microsclerotium Formation

The *V. dahliae* used in this study (At13) was isolated from *A. truncatum* Bunge in Shandong Province, China. All of the *V. dahliae* strains were preserved in 50% glycerol at −80 °C. To observe the morphology of mycelium and spores of these strains, the *V. dahliae* was cultivated on potato dextrose agar (PDA) (potato, 200 g/L; glucose, 20 g/L; agar, 20 g/L) medium at 25 °C or on the shaking incubator in complete medium (CM) (yeast extract, 6 g/L; casein acids hydrolysate, 6 g/mL; sucrose, 10 g/L) at 25 °C.

To observe spore numbers produced by these strains, mycelium blocks of each strain were collected using a punch and were shaken thoroughly in sterile water, and the number of spores was counted on a blood cell counting plate. To observe the morphology of hyphae and spores of each strain, hyphae were collected after 5 days of growth on hydrophobic glass slides, and spores and hyphal morphology were observed microscopically (Scale bar = 20 µm). To investigate the effect of terbinafine on the growth of *V. dahliae*, 0.02 μg/mL terbinafine was added to PDA medium, strains were grown at 25 °C for 10 days, and plates were photographed. At least 3 dishes of each fungal strain were inoculated.

To elucidate the phenotypic effects of growth on different carbon sources, 2 mm × 2 mm fungal plugs were excised with a surgical knife from colonies of the wild-type, deletion mutant, and complemented strains. They were then placed centrally on 6 cm diameter Czapek plates supplemented with different carbon sources (NaNO_3_, 2 g/L; K_2_HPO_4_, 1 g/L; MgSO_4_·7H_2_O, 0.5 g/L; KCl, 0.5 g/L; FeSO_4_, 0.01 g/L; sucrose, 30 g/L or starch, 17 g/L or levulose, 17 g/L or galactose, 17 g/L). At least 3 dishes of each fungal strain were inoculated. After ten days of cultivation at 25 °C, the diameters of the fungal colonies were measured using a Vernier caliper.

To investigate the effect of *VdKeR1* on melanin synthesis and microsclerotia formation in *V. dahliae*, uniform-sized agar plugs were cut from colonies of the wild-type, deletion mutant, and complemented strains. These plugs were then placed on PDA and V8 culture media (Campbell, Camden, NJ, USA). After 14 days of cultivation at 25 °C, plates were photographed. At the same time, spore suspensions of the wild-type, deletion mutant, and complemented strains were adjusted to 1 × 10^6^ conidia/mL using the hemocytometer (Sangon Biotech), and then were spread on basal agar modified medium (BMM) covered with cellophane. After two weeks of cultivation at 25 °C, microsclerotia on cellophane were observed and photographed using a Zeiss stereo microscope (Zeiss) (scale bar = 50 µm).

### 2.5. Role of VdKeR1 in Osmotic, Oxidative, Cell Wall, and Terbinafine Stress

To investigate whether *VdKeR1* affects the sensitivity of *V. dahliae* to osmotic, oxidative, and cell wall stress, 2 mm × 2 mm fungal plugs of the wild-type, deletion mutant, and complemented strains were grown on Czapek medium containing 0.5 M/L NaCl or 1 M/L sorbitol or 2.5 mM/L H_2_O_2_ or 0.014% SDS or 200 μg/mL Congo red or 20 μg/mL CFW. At least 3 dishes of each fungal strain were inoculated. After 10 days of incubation at 25 °C the colony diameters were measured, and the inhibition rate was calculated.
Inhibition rate (%) = (colony diameter on Czapek medium−colony diameter on stress medium)/colony diameter on Czapek medium × 100%.

To investigate the inhibitory effect of terbinafine on *V. dahliae*, 2 mm × 2 mm mycelium blocks of wild-type, deletion mutant, and complemented strains were inoculated on the control and terbinafine PDA plates. Then strains were cultivated at 25 °C for 10 days. After 10 days, the colony diameters were measured with a Vernier caliper, and the fungal inhibition rate of each strain was calculated.
Inhibition rate (%) = (colony diameter on PDA medium−colony diameter on stress medium)/colony diameter on PDA medium × 100%.

### 2.6. Penetration and Pathogenicity Analysis

To analyze the role of *VdKeR1* in the penetration of *V. dahliae*, sterilized and dried cellophane was placed on PDA medium, followed by the placement of a 2 mm × 2 mm fungal plug in the center. The fungi were then cultivated at 25 °C for 3, 4, and 5 days. After this time, the cellophane was removed and cultivation continued for a further two days. Photography and observation were conducted both before and after removing the cellophane.

To analyze the role of *VdKeR1* in the pathogenicity of *V. dahliae*, the fungi were cultured on PDA medium for seven days. The fungal plugs were then transferred to CM medium and shaken for two days to obtain spore suspension. After filtration and centrifugation, the spores were washed, and the concentration of WT, mutant and complemented strains spore suspension was adjusted to 5 × 10^6^ conidia/mL using the hemocytometer. Susceptible cotton seedlings (*Gossypium hirsutum* cv. Junmian 1) were grown in substrate composed of nutrient soil and vermiculite in a 1:1 ratio for three weeks. The roots were then removed from the substrate and soaked in a spore suspension of adjusted concentration using a root-dip inoculation method [46]. The experiment included 20 cotton and 5 maple seedlings, with water and wild-type strain used as negative and positive controls, respectively. Three repeated toxicity analyses were conducted using the aforementioned method after obtaining the complemented strains of *VdKeR1*. On the 18th day after cotton inoculation and the 21st after maple inoculation, we observed the disease phenotype and took photographs. Subsequently, cotton stem samples were collected for fungal re-isolation and biomass analyses.

### 2.7. RT-qPCR for Analysis of Relative Gene Expression and qPCR for Fungal Biomass

To detect the expression level of melanin-related genes by *VdKeR1*, total RNA Isolation kit (Vazyme, Nanjing, China) was used to extract RNA from mycelium of At13, mutants, and complemented strains. cDNA was obtained by reverse transcription using One-Step gDNA Removal and cDNA Synthesis Super Mix kit (Trans Gen Biotech, Beijing, China). RT-qPCR was performed under the following condition: an initial denaturation at 95 °C for 20 s, followed by 40 cycles of 95 °C for 15 s, 60 °C for 20 s, and 72 °C for 20 s.

To investigate expression level of genes related to microsclerotia and melanin synthesis, *VdEF-1 α* gene of *V. dahliae* was used as internal control, with WT as control. To detect the fungal biomass at the root-stem junction, *VdEF-1 α* gene of *V. dahliae* and *18S* gene of cotton and maple were used as internal controls, with WT as control. To detect the expression level of *sqe* gene in the absence and presence of terbinafine, *VdEF-1 α* gene of *V. dahliae* was used as an internal control, with WT as control. The control group and 0.02 μg/mL terbinafine (Sangon Biotech) group were tested. The relative transcription levels of the above genes in different strains were determined using the 2^−ΔΔCT^ method [47]. The primers used in RT-qPCR are listed in Appendix A. Each treatment was replicated three times, and the experiment was performed three times.

### 2.8. Extraction and Content Determination of Squalene and Ergosterol

WT and complemented strains were grown on PDA plates coated with glass paper at 25 °C for 4 days. To prepare the samples, 1 g of glass paper containing mycelium was weighed into a test tube. Squalene content was determined by the method of Mengfan [48]. A total of 15% (weight/volume) potassium hydroxide methanol/H_2_O (4:1, volume ratio) was added to the glass paper and saponified in a water bath at 60 °C for 1.5 h. After cooling to room temperature, the unsaponifiable portion was extracted with 2 mL of n-hexane. The supernatant was filtered through a 0.45 μm filter and transferred to a brown chromatography vial for liquid chromatography. Using squalene as the external standard, the content of squalene (mg/L) was determined by liquid chromatography. The chromatographic determination conditions were high-performance liquid chromatography (Agilent 1260); mobile phase: acetonitrile/methanol (1:1, volume ratio); flow rate: 1.0 mL/min; column temperature: 30 °C and detection wavelength was 210 nm.

Ergosterol content was determined as described by Yaliang [49]. A total of 6 mL of 50% KOH and 4 mL of 95% ethanol were added and saponified in an 80 °C water bath for 1.5 h. Then 2.5 mL of 95% ethanol was added and saponified for another 1 h. After cooling to room temperature, 10 mL of ether was added and shaken for 15 min. The mixture was allowed to stand at room temperature for 2 h. After stratification, the supernatant was filtered through a 0.45 μm filter head and transferred to the brown chromatography vial for liquid chromatography. Using ergosterol as the external standard, the content of ergosterol (mg/L) was determined by liquid chromatography. The chromatographic determination conditions were high-performance liquid chromatography (Agilent 1260); mobile phase: methanol 100%; flow rate: 1 mL/min; column temperature: 30 °C; detection wavelength: 271 nm.

Each experiment was repeated three times, and each sample was injected twice.

### 2.9. Statistical Analysis

The data were visualized with bar and line charts using GraphPad Prism 8.0 (GraphPad Inc., Beijing, China). The values displayed in the bar charts represented the means of three replicates. The error bars in the bar charts indicated the standard deviations (SD) calculated from the three replicates. Asterisks denoted significant differences between the mutants and WT, as well as between the mutants and the complemented strains, as determined by one-way analysis of variance (ANOVA) (** *p* < 0.01). All statistical analyses were conducted with SPSS version 25 (SPSS Inc., Chicago, IL, USA).

## 3. Results and Analysis

### 3.1. Identification of the Kelch Repeat Domain-Containing Protein VdKeR1 in V. dahliae

Previous studies have shown that the polyketide synthase 9 (PKS9)-included regions are conserved among filamentous fungi [35]. Therefore, we conducted an identification study on the hypothetical protein DK185_4252 (VdLs.17 VDAG_08647) belonging to the VdPKS9 gene cluster (Figure 1A). The full-length sequence with 1833 bp was obtained by amplification of the genome of the *V. dahliae* strain At13. Sequence analysis revealed that this gene lacks introns and encodes 610 amino acids. Furthermore, according to the prediction and analysis carried out using the SMART and InterPro online tools, this sequence contains six Kelch repeat motifs located between amino acids 70 and 579. (Figure 1B). As a result, the protein was named VdKeR1. Models of the secondary and tertiary structure were constructed, revealing that the six Kelch repeat motifs formed a six-bladed propeller structure. Each Kelch repeat consisted of six antiparallel β-sheets, which together formed a β-propeller secondary structure in *V. dahliae.* (Figure 1C). Homologous sequences of VdKeR1 were collected from other filamentous fungi in the NCBI database, and a phylogenetic tree was constructed. Phylogenetic analysis showed that VdKeR1 had the highest sequence homology with sequences from *Verticillium* spp., followed by *Colletotrichum* spp. It was also found that VdKeR1 was conserved among filamentous fungi (Figure 1D). VdKeR1-GFP strains were obtained as described above and analyzed using fluorescence microscopy. The strains exhibited high fluorescence intensity and no significant change in colony morphology. The GFP-tagged VdKeR1 showed an even distribution in the cytoplasm of germinating spores and hyphae (Figure 1E).

### 3.2. VdKeR1 Is Involved in the Growth and Development of V. dahliae

To investigate the function of the *VdKeR1* gene, we obtained *VdKeR1* deletion mutants (Δ*VdKeR1*) through the homologous recombination method and complemented strains (EC_*VdKeR1*) through the random insertion method as shown in Appendix A. The PCR verification of the transformants is presented in Appendix A. Details of the primers used can be found in Appendix A.

The WT, Δ*VdKeR1*, and EC_*VdKeR1* strains were cultured on PDA plates. After 10 days of cultivation at 25 °C, the Δ*VdKeR1* strains exhibited significantly slower growth compared to the WT and EC_*VdKeR1* (Figure 2A,B) strains. The Δ*VdKeR1* strains showed a significant increase in aerial mycelia and had raised colonies, while the WT and EC_*VdKeR1* strains had relatively flat colonies due to less mycelial growth (Figure 2A).

As with most filamentous fungal pathogens, asexual conidia play a crucial role in the infection and disease progression of *V. dahliae* [50]. To investigate the role of *VdKeR1* in conidiation, we counted the number of conidia produced by each strain. The results showed that the Δ*VdKeR1* strains produced significantly fewer conidia compared to the WT and EC_*VdKeR1* (Figure 2C) strains. The WT and EC_*VdKeR1* strains produced approximately 3 × 10^5^ conidia/mL, while the Δ*VdKeR1* strains produced approximately 2 × 10^4^ conidia/mL after being cultured in CM liquid medium for 2 days (Figure 2D).

It is worth noting that carbon is an essential nutrient for fungal growth and development [51]. When different carbon sources (such as sucrose, starch, fructose, and D-galactose) were added to the Czapek medium, the growth of each strain was assessed by comparing colony diameters. There was no difference in growth between the WT, Δ*VdKeR1*, and EC_*VdKeR1* strains when D-galactose was used as the carbon source. However, when sucrose, starch, or fructose were used as the carbon source, the Δ*VdKeR1* strains showed a significant reduction in colony diameter compared to the WT and EC_*VdKeR1* strains (Figure 2E,F). The comparisons were made between strains grown on the same carbon source.

These results indicated that *VdKeR1* was involved in the growth and development process of hyphae and conidia in *V. dahliae*.

### 3.3. VdKeR1 Was Related to Osmotic, Oxidative, and SDS-Induced Cell Wall Stress Processes

Fungal cell walls and membranes are crucial barriers for fungi, providing protection against adverse external factors [52]. The Δ*VdKeR1* strains showed greater tolerance to the osmotic stress factors of 0.5 M/L NaCl and 1 M/L sorbitol compared to the WT and EC_*VdKeR1* strains. This indicated that *VdKeR1* was involved in regulating the osmotic stress response in *V. dahliae*. In contrast, the Δ*VdKeR1* strains exhibited greater sensitivity when exposed to 2.5 mM/L H_2_O_2_ than the WT and EC_*VdKeR1* strains (Figure 3A,B). This suggested that *VdKeR1* acted as a positive regulator in the oxidative stress response induced by H_2_O_2_ in *V. dahliae*

The results indicated that the Δ*VdKeR1* strains exhibited significant differences compared to the WT and EC_*VdKeR1* strains when exposed to the cell wall stress factors of 0.014% SDS, 200 μg/mL Congo red, and 20 μg/mL CFW. Specifically, the Δ*VdKeR1* strains were found to be more sensitive to 0.014% SDS and more tolerant to 200 μg/mL Congo red and 20 μg/mL CFW (Figure 3C,D). This suggested that *VdKeR1* played an important role in the response to SDS-induced cell wall stress.

### 3.4. VdKeR1 Affects Melanin Synthesis and Formation of Microsclerotia

Melanin is essential for the development and functionality of fully equipped microsclerotia in *V. dahliae* [53]. Melanin and microsclerotia can aid *V. dahliae* in resisting ultraviolet radiation and extreme temperatures. Although melanin itself is not a virulence factor of *V. dahliae* [54], the metabolic pathways that regulate melanin biosynthesis and microsclerotia formation can affect the virulence of *V. dahliae*. To investigate the role of *VdKeR1* in these physiological processes in *V. dahliae*, we cultivated the WT, Δ*VdKeR1*, and EC_*VdKeR1* strains separately on PDA and V8 media at 25 °C for two weeks. We observed that the Δ*VdKeR1* strains exhibited a reduction in melanin production, while the WT and EC_*VdKeR1* strains showed significant melanization (Figure 4A). The suspension concentrations of conidia from the WT, Δ*VdKeR1* and EC_*VdKeR1* strains were adjusted to approximately 5 × 10^6^ conidia /mL. They were then spread onto microsclerotia induction medium (BMM medium) covered with cellophane. After incubation at 25 °C for two weeks, observations were made using a stereo microscope. It was observed that both the WT and EC_*VdKeR1* strains were able to form well-pigmented microsclerotia, while the Δ*VdKeR1* strains failed to form mature microsclerotia (Figure 4B,C). The presence of melanin in the WT indicates the presence of microsclerotia [50]. The RT-qPCR expression analyses of genes related to melanin synthesis in *V. dahliae* revealed that the expression levels of these genes were significantly downregulated in the Δ*VdKeR1* strains, while the expression levels in the EC_*VdKeR1* strains were restored to the level of the WT (Figure 4D). These results provided evidence that *VdKeR1* was involved in the expression of genes related to melanin synthesis, thus playing a crucial role in melanin synthesis in *V. dahliae*.

### 3.5. VdKeR1 Deletion Can Delay Penetration Ability of V. dahliae

This study aimed to investigate the role of *VdKeR1* in early *V. dahliae* infection of plants. To achieve this, mycelial blocks from the WT, Δ*VdKeR1*, and EC_*VdKeR1* strains were cut and placed on PDA medium covered with the cellophane to analyze their penetration ability. After incubating at 25 °C for 3, 4, and 5 days, the cellophane membranes were removed, and on the seventh day, the penetration of mycelium through the cellophane membranes was observed (Figure 5). The study found that the mycelium of the Δ*VdKeR1* strains had a significantly slower penetration rate through the cellophane membranes compared to the WT. On the third day, the mycelia of the Δ*VdKeR1* strains failed to penetrate the cellophane membranes, while the mycelia of the WT and EC_*VdKeR1* strains were able to do so. Additionally, the EC_*VdKeR1* strains regained the same penetration speed as the WT. These results suggested that *VdKeR1* played a role in the penetration of *V. dahliae* into host plants.

### 3.6. VdKeR1 Plays a Critical Role in Pathogenicity

To investigate the effect of *VdKeR1* on the pathogenicity in cotton and maple seedlings, spore suspensions of the WT, Δ*VdKeR1*, and EC_*VdKeR1* strains were inoculated separately onto cotton and maple. The results showed that the pathogenicity of the Δ*VdKeR1* strains in cotton and maple was significantly decreased (Figure 6A,D). Disease index statistics for cotton and maple showed that the incidence rate and the incidence degree of the Δ*VdKeR1* strains were significantly lower than those of the WT and EC_*VdKeR1* strains (Figure 6B,E). The biomass of the Δ*VdKeR1* strains detected in inoculated cotton and maple was also significantly reduced. The pathogenicity and biomass detected in the EC_*VdKeR1* strains were restored to the WT level (Figure 6C,F). These results suggested that *VdKeR1* played a role in the pathogenic process of *V. dahliae.*

### 3.7. VdKeR1 Positively Regulates the Activity of Squalene Epoxidase

Terbinafine inhibits the target site for squalene epoxidase (SQE) in the ergosterol synthesis pathway. This disruption of the fungal membrane structure leads to a significant decrease in the synthesis and conversion functions of phospholipids and proteins, resulting in reduced membrane function and death of the fungi [55]. The aim of this investigation was to determine whether the pathogenic mechanism of *V. dahliae* was related to the targeting of terbinafine by the *VdKeR1* gene. To achieve this, we added 0.02 μg/mL terbinafine to PDA and observed the growth changes of the WT, Δ*VdKeR1*, and EC_*VdKeR1* strains after 10 days. The results showed that the Δ*VdKeR1* strains exhibited a certain degree of resistance to terbinafine compared to the WT and EC_*VdKeR1* strains, while the WT and EC_*VdKeR1* strains were highly sensitive, with an inhibition rate of around 42% (Figure 7B). The comparison was between the bottom and top rows of Figure 7A. Analysis of the expression level of *sqe* showed that in the absence of terbinafine, the expression level of *sqe* in the Δ*VdKeR1* strains was significantly higher than that in the WT and EC_*VdKeR1* strains. However, in the group with added terbinafine, there was no significant change in the Δ*VdKeR1* strains, but the expression levels of the WT and EC_*VdKeR1* strains increased significantly, by 2.4 times the original (Figure 7C). These findings indicated that *VdKeR1* played a role in the fungal response to terbinafine, with the absence of *VdKeR1* conferring increased resistance. *VdKeR1* may regulate the expression of squalene epoxidase, particularly in response to terbinafine. The qPCR results indicated that *VdKeR1* may regulate ergosterol biosynthesis by modulating the expression of the *sqe* gene.

### 3.8. VdKeR1 Affects the Synthesis of Squalene and Ergosterol

Terbinafine is a drug used to treat fungal infections by targeting squalene epoxidase, an enzyme that plays a crucial role in the sterol biosynthetic pathway of eukaryotes [56]. Furthermore, ergosterol is an end-product in the squalene synthesis pathway and is an important component for maintaining cell membrane stability. Its biosynthesis is closely related to the growth and pathogenicity of fungi [57]. The observed expression levels of *sqe* suggested that *VdKeR1* might regulate the synthesis of squalene and ergosterol in this pathway. To investigate this, we measured the content of squalene and ergosterol. These results showed that the squalene content in the control groups was significantly higher in the Δ*VdKeR1* strains than in the WT and EC_*VdKeR1* strains. After the addition of 0.02 μg/mL terbinafine, there was no significant change in squalene content in the Δ*VdKeR1* strains. However, the squalene content in the WT and EC_*VdKeR1* strains significantly increased, roughly tenfold compared to the original level (Figure 8B–D). At the same time, the ergosterol content in the Δ*VdKeR1* strains was significantly lower than that in the WT and EC_*VdKeR1* strains in the control groups. Following the addition of 0.02 μg/mL terbinafine, the ergosterol content decreased in the Δ*VdKeR1* strains. However, in the WT and EC_*VdKeR1* strains, the ergosterol content decreased to approximately 57 times lower than that in the control groups (Figure 8F–H).

These experimental results suggested that *VdKeR1* positively regulates the activity of squalene epoxidase and mediates the synthesis of squalene and ergosterol. This implies that the absence of the *VdKeR1* gene leads to the accumulation of squalene and the deficiency of ergosterol in *V. dahliae*.

## 4. Discussion

This study identified the protein VdKeR1 (DK185_4252). The *VdKeR1* gene belongs to the *VdPKS9* gene cluster, which regulates the formation of microconidia and the growth of hyphae in *V. dahliae*. The protein VdKeR1 was found to contain Kelch repeat motifs, which were highly conserved in filamentous fungi. The role of *VdKeR1* in growth, development, spore production, and virulence of *V. dahliae* was analyzed. This research showed that *VdKeR1* positively regulated the activity of squalene epoxidase in *V. dahliae*, thereby mediating the synthesis of squalene and ergosterol.

In fungi, most proteins that contain Kelch domains are associated with hyphal growth and development [58]. The Kelch protein structure encoded by Tea1 in *Schizosaccharomyces pombe* comprises a β-propeller structure formed by Kelch repeat motifs at the N-terminus. Its function involves binding to microtubule end proteins, causing microtubules to aggregate at the tip of the hyphae, promoting the polar growth of fungal hyphae [22]. The *Clakel2*-encoded Kelch protein structure in *C. lagenarium* is located at the N-terminus. This protein is localized in the polarized regions of growing hyphae and germ tubes. Knocking out this gene can affect the formation of appressoria [59]. The Kelch repeat motifs spanned basically the entire polypeptide sequence of the VdKeR1 protein. Knocking out the *VdKeR1* gene resulted in a decreased growth rate of the Δ*VdKeR1* strains, an increase in aerial hyphae, and a decrease in spore production. No correlation was found between the hyphae and spores with polar growth. Some proteins containing Kelch repeat motifs, such as the Kelch proteins found in *Dictyostelium dendroides* and *Hypomyces rosellus*, may have a structure similar to galactose oxidase [60]. However, according to protein structure predictions in our study, it appeared that *VdKeR1* did not contain a galactose oxidase structure in *V. dahliae*. Further research is required to investigate the role of this structure in proteins containing Kelch repeat motifs.

The microsclerotia of *V. dahliae* are the primary source of infection for this pathogen. The production of melanin is crucial for the development of microconidia. Pigments produced by the pathogen help it withstand adverse environmental conditions, enabling survival and protecting the pathogen from host oxidative stress during invasion [61]. In addition, DHN melanin is considered a determining factor for stress resistance, pathogen–host interactions, pathogenicity, and virulence [62,63]. When comparing the Δ*VdKeR1* strains to the WT and EC_*VdKeR1* strains, the Δ*VdKeR1* strains produced significantly less melanin and failed to form mature microconidia. However, the deletion of the *CoKEL1* gene in *C. orbiculare* forms irregularly shaped appressoria but does not affect the melanin and formation of appressoria [64]. This may be due to the functional differentiation of Kelch repeat motifs in different organisms. Genes related to melanin and microconidia synthesis in *V. dahliae* include *VdPKS1*, *VdLac1*, and *VdCmr1* [54,65]. RT-qPCR expression analyses of genes related to melanin synthesis in *V. dahliae* revealed that the expression levels of these genes were significantly downregulated in the Δ*VdKeR1* strains. Although melanin is not a virulence factor of *V. dahliae* [54], it is crucial for the development and functionality of fully equipped microsclerotia in *V. dahliae* [53]. The metabolic pathways regulating melanin biosynthesis and microsclerotia formation could affect the virulence of *V. dahliae.*

The biological functions of the *VdKeR1* protein in response to external osmotic stress, oxidative stress, and cell wall stress were further explored. The results showed that, after adding an oxidative stress factor, the Δ*VdKeR1* strains significantly differed from the WT and EC_ *VdKeR1* strains. This indicated the involvement of *VdKeR1* in the oxidative stress responses of *V. dahliae*. Considering the higher sensitivity of the Δ*VdKeR1* mutants to SDS and their higher tolerance to CR and CFW, we concluded that *VdKeR1* played an important role in the response to SDS-induced cell wall stress. Research has indicated that the inhibition mechanism of SDS on fungal cell walls involves disrupting oxidative stress metabolism and lipid metabolism [66]. But the mechanisms for CR and CFW inhibition are related to inhibiting chitin synthesis [67]. These different mechanisms may explain the contrasting results observed for SDS versus CR and CFW. The VdKeR1 protein was located in the hyphae and cytoplasm of conidia, which was markedly different from the localization of most Kelch domain proteins at the tips of hyphae and conidia.

During the investigation of the pathogenic mechanism of *VdKeR1*, it was observed that the Δ*VdKeR1* strains exhibited increased tolerance to the addition of 0.02 μg/mL terbinafine compared to the WT and EC_*VdKeR1* strains. The results of the relative expression levels of *sqe* indicated that terbinafine might affect the regulatory molecules of the downstream biosynthesis pathway of squalene. Terbinafine inhibits the activity of squalene epoxidase, which might counteract the regulation of the squalene epoxidase gene and increase its expression. This was consistent with previous studies on the expression of the squalene epoxidase gene in *Thraustochytrids*. Those studies indicate that terbinafine not only reduces the activity of squalene epoxidase, but also increases the expression level of the *sqe* gene [48]. Previous study finds that terbinafine acts on the target site of squalene epoxidase in the ergosterol synthesis pathway, affecting the enzyme’s activity and inhibiting the further oxidation of lanosterol, thus blocking the synthesis of ergosterol [13]. In our study, the squalene content in the Δ*VdKeR1* strains increased significantly compared to the WT and EC_*VdKeR1* strains, even without the addition of terbinafine. However, upon the addition of terbinafine, the Δ*VdKeR1* strains showed little change in comparison to the WT and EC_*VdKeR1* strains. In contrast, significant changes were observed in the WT and EC_*VdKeR1* strains, with squalene content increasing to around 10 times the original level and ergosterol content decreasing by around 57 times. In previous research, the deletion of the *SsCI80130* gene encoding squalene epoxidase in *S. scitamineum* led to an increase in squalene content and a decrease in ergosterol content [15].

All experimental results indicated that *VdKeR1* not only regulated the growth and development, conidial production, melanin synthesis, and microsclerotia formation of *V. dahliae*, but also participated in cell membrane osmotic and cell wall stress responses. Additionally, deletion of *VdKeR1* could delay the penetration ability of fungal hyphae into host, mediating the pathogenicity of *V. dahliae*. Furthermore, we found that the absence of *VdKeR1* led to the accumulation of squalene, inhibited squalene epoxidase activity, disrupted the ergosterol synthesis pathway, and caused a deficiency in ergosterol while increasing the expression of the *sqe* gene.

## 5. Conclusions

In conclusion, these findings helped to understand the biological role of *VdKeR1* in *V. dahliae* and broadened our knowledge of the diversity of functional proteins within fungi. However, whether the reduction in virulence observed in △*VdKeR1* was associated with disruptions in the ergosterol synthesis pathway remains to be further investigated. Future research can focus on the squalene epoxidase and ergosterol synthesis pathway in *V. dahliae*. Moreover, it can further elucidate the pathogenic mechanisms involving *VdKeR1*. It can provide theoretical and experimental foundations for developing new inhibitors to control *Verticillium* wilt.

## Figures and Tables

**Figure 1 jof-10-00643-f001:**
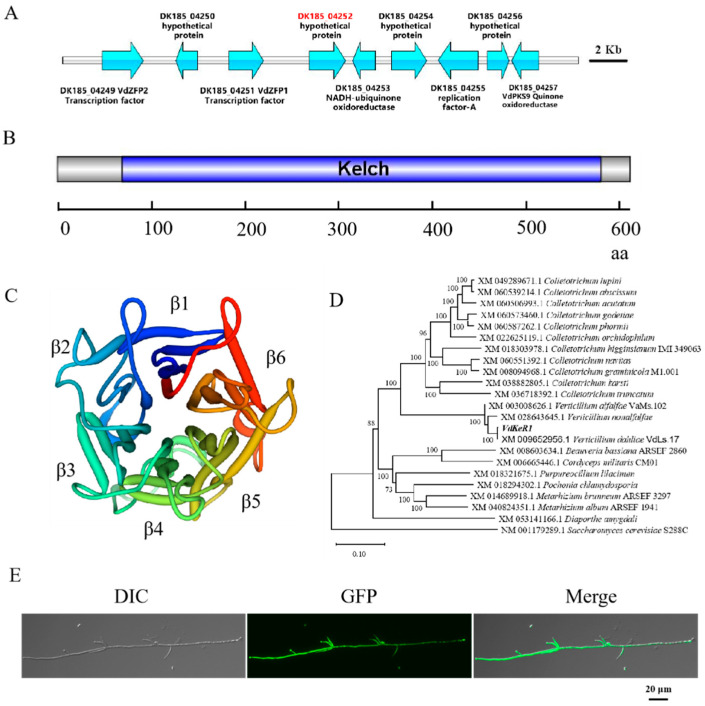
Identification and subcellular localization of VdKeR1 in *V. dahliae*. (**A**) Origin of VdKeR1. Red indicates DK185-0452 (VdKeR1) position on the *VdPKS9* gene cluster. (**B**) Prediction of VdKeR1 structural domain. The Kelch structural domain is highlighted in blue. Numbers of corresponding amino acids are shown below scale bar. (**C**) Secondary structure VdKeR1 protein. Numerical numbers represent coding sequences. β1–β6 refer to the β-sheets formed by the Kelch repeat motifs. (**D**) Phylogenetic tree analysis of the proteins encoded by the *VdKeR1* gene. Arabic numerals at the edges of the tree nodes indicate the confidence levels of the phylogenetic tree (using 1000 bootstrap replicates). Protein encoded by *VdkeR1* is boldfaced. (**E**) Subcellular localization of VdKeR1 protein. The VdKeR1-GFP fusion fragment was inserted into the genome of WT strain At13. The localization of VdKeR1-GFP in hyphae and conidia was observed by fluorescence microscopy.

**Figure 2 jof-10-00643-f002:**
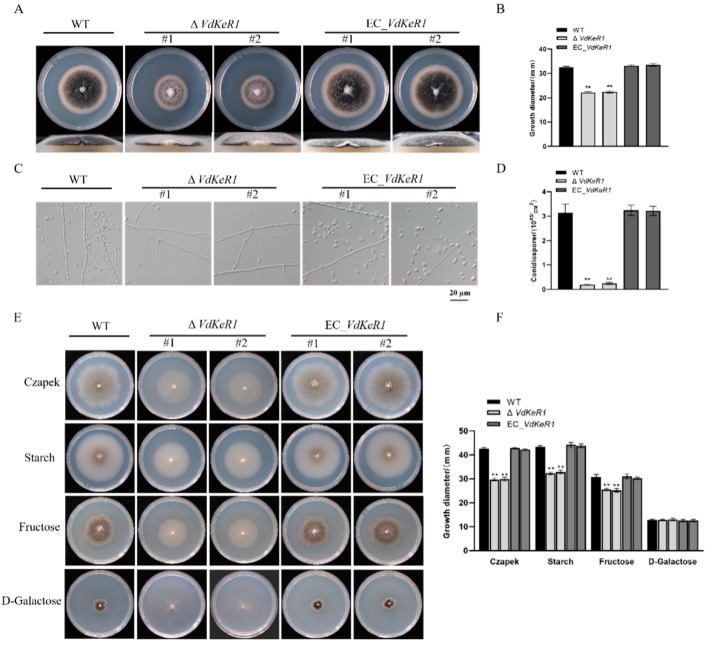
The effect of *VdKeR1* gene on the normal growth and the production of conidia in *V. dahliae*. (**A**) The mycelial growth of each strain. Each strain was inoculated on a PDA plate, grown at 25 °C for 10 days, and photographed. (**B**) Measures of the colony diameters of the Petri dishes in Panel A using a Vernier caliper. Mean diameters calculated from three colonies. (**C**) The morphology of the hyphae and spores of each strain. (Scale bar = 20 µm). (**D**) Spore production statistics. Mycelium blocks of each strain were collected and spore numbers were determined as described in M and M. (**E**) Growth of each strain under different carbon source conditions. (**F**) Measures of the colony diameters of the Petri dishes in Panel E using a Vernier caliper. Mean diameters calculated from three colonies. Symbols #1 and #2 represent two replicate strains of the mutant and complemented strains; the error lines represent the standard deviation of at least three independent measurements; and significance was analyzed by one-way ANOVA. ** Δ*VdKeR1* was significantly different from WT and complemented strains (*p* < 0.01).

**Figure 3 jof-10-00643-f003:**
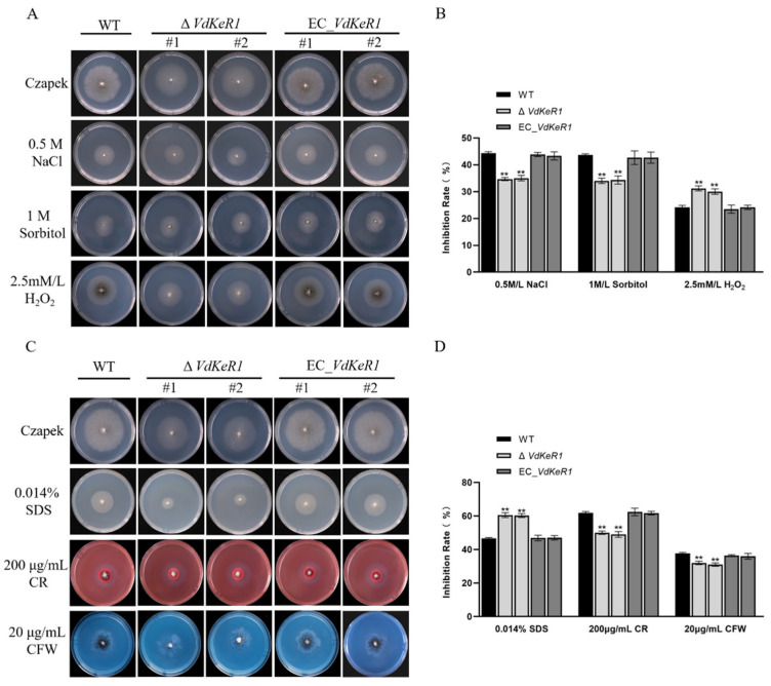
Effect of *VdKeR1* gene on sensitivity to osmotic stress, oxidative stress, and cell wall stress in *V. dahliae*. (**A**) Growth of each strain under different osmotic and oxidative stress factors. (**B**) The inhibition rate of each strain under different osmotic and oxidative stress factors. (**C**) The growth of each strain under different cell wall stress chemicals. (**D**) The inhibition rate of each strain under different cell wall stress chemicals. ** Δ*VdKeR1* was significantly different from WT and complemented strains (*p* < 0.01).

**Figure 4 jof-10-00643-f004:**
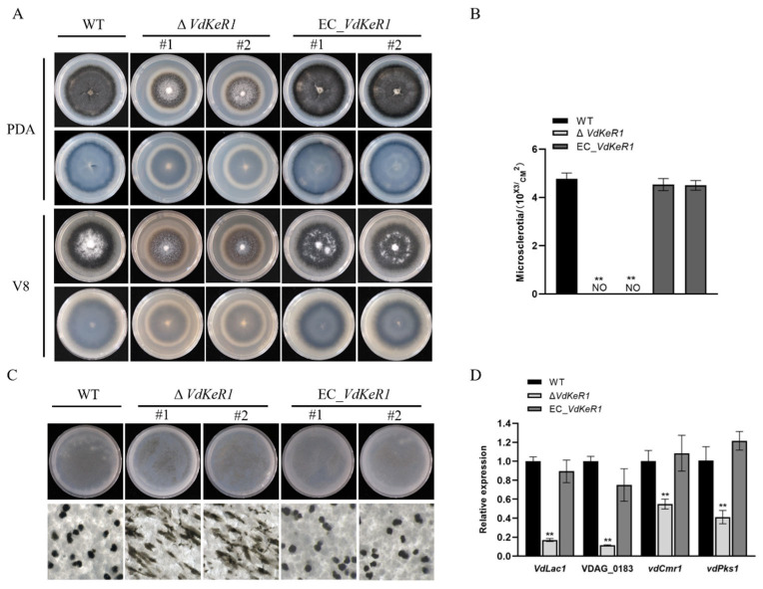
Effect of *VdKeR1* gene on melanin synthesis and microsclerotia formation in *V. dahliae*. (**A**) Melanin synthesis on V8 and PDA media. (**B**) Formation of microsclerotia. The spore suspension of each strain was spread on BMM medium covered with glass paper. (scale bar = 50 µm) (**C**) Micronucleus count statistics. The photographs in the bottom row of (**C**) were entered into Image J 1.5 software to obtain the number of particles per unit area. (**D**) Expression levels of genes related to microsclerotia formation and melanin synthesis. Gene expression analysis method as described in M and M. ** Δ*VdKeR1* was significantly different from WT and complemented strains (*p* < 0.01).

**Figure 5 jof-10-00643-f005:**
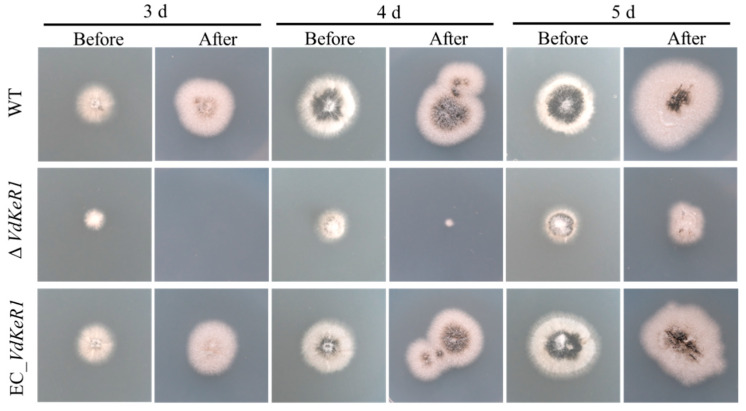
Effect of *VdKeR1* gene on the penetration ability of *V. dahliae*. Inoculated each strain in the center of a cellophane membrane on the surface of PDA, took photographs and removed the cellophane membrane on 3rd, 4th, and 5th day, respectively, and observed the penetration of the strains on 7th day. Prepared at least three dishes for each strain. “Before” referred to the photos taken at 3rd, 4th, and 5th day when the cellophane had not been removed. “After” referred to the photos taken after removing the cellophane and allowing *V dahliae* to grow up to 7 days.

**Figure 6 jof-10-00643-f006:**
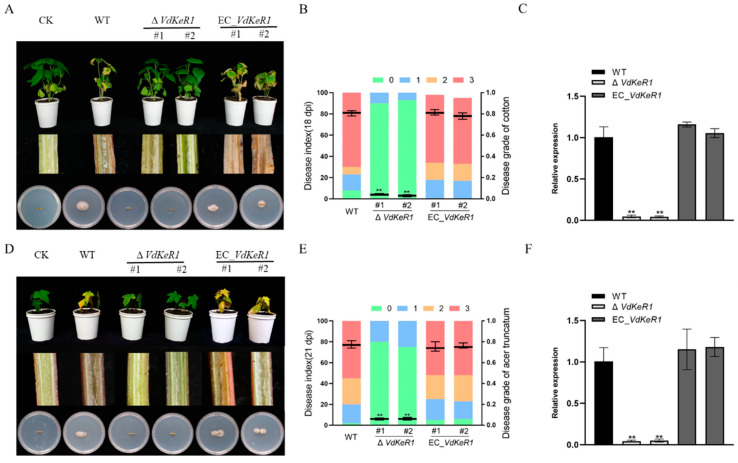
Effect of *VdKeR1* gene on the pathogenic process of *V. dahliae*. (**A**–**C**) Pathogenic effect of *VdKeR1* on cotton. (**D**–**F**) Pathogenic effect of *VdKeR1* on maple. (**A**,**D**) Results of the infection on phenotype. The phenotype of cotton and maple seedlings was recorded at 18 dpi and 21 dpi, respectively. The top part of the pictures shows the symptoms of the disease; the middle part shows longitudinal sectional images of the stems after inoculation, to observe the discoloration of the vascular bundle; and the bottom part shows the fungal growth on the stem sections inoculated on the plates after 7 days. (**B**,**E**) Statistics of disease index, grading, and incidence rate calculation of seedlings infected with each strain. (**C**,**F**) qPCR detection of fungal biomass at the root–stem junction. Gene expression analysis method as described in M and M. ** Δ*VdKeR1* was significantly different from WT and complemented strains (*p* < 0.01).

**Figure 7 jof-10-00643-f007:**
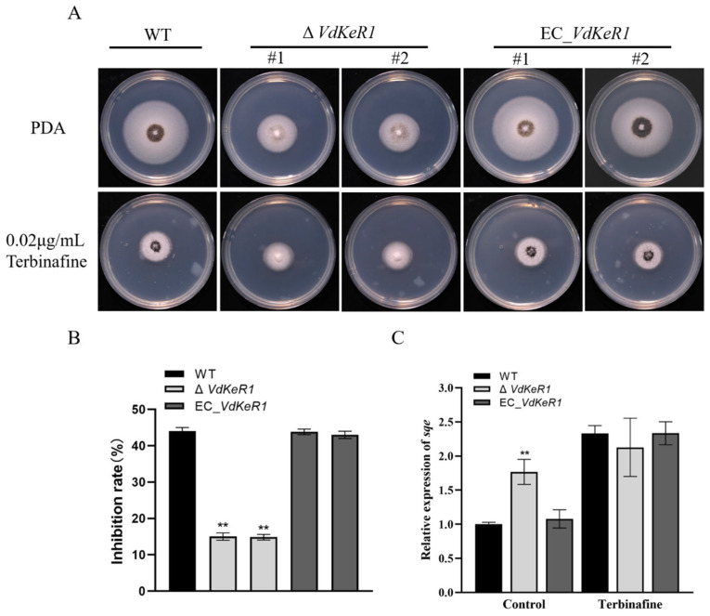
Response of WT, deletion mutant, and complemented strains l to terbinafine. (**A**) Growth of *V. dahliae* strains on PDA plates with and without terbinafine. (**B**) The inhibition rate of each strain under 0.02 μg/mL terbinafine. (**C**) qPCR was used to detect the expression levels of *sqe* gene in the absence and presence of terbinafine. Gene expression analysis method as described in M and M. ** Δ*VdKeR1* was significantly different from WT and complemented strains (*p* < 0.01).

**Figure 8 jof-10-00643-f008:**
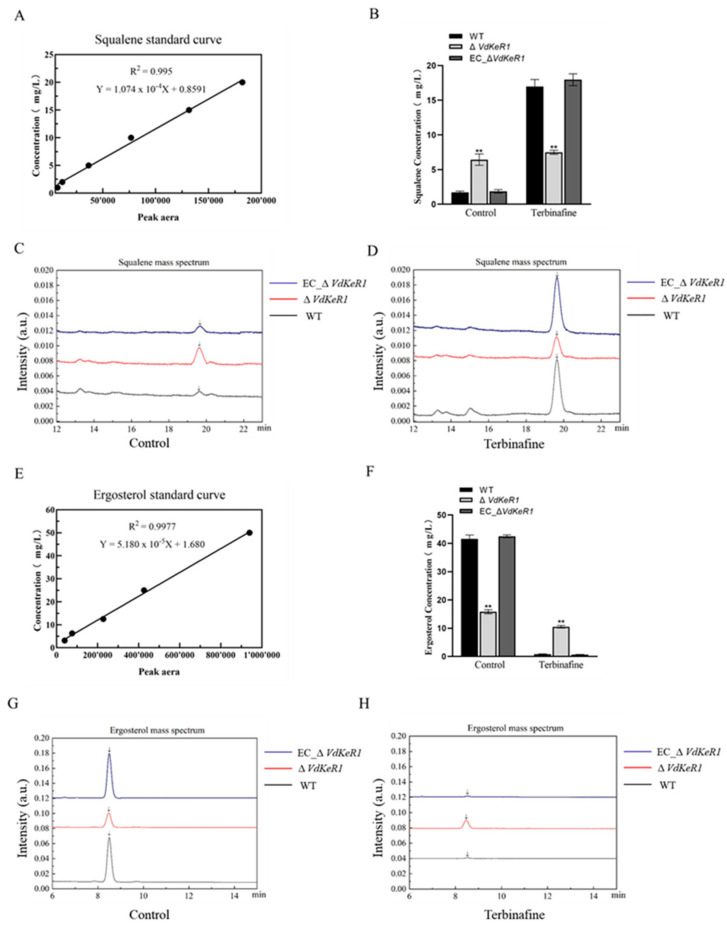
Effect of *VdKeR1* gene on squalene and ergosterol content in *V. dahliae*. (**A,E**) The standard curves of squalene and ergosterol. (**B**,**F**) Each strain was grown at 25 °C for 4 d under control and 0.02 μg/mL terbinafine before squalene and ergosterol extraction. At least 3 dishes of each fungal strain were inoculated. (**C**,**G**) Squalene and ergosterol mass spectra of each strain in the control group. (**D**,**H**) Squalene and ergosterol mass spectra of each strain in the 0.02 μg/mL terbinafine group. ** Δ*VdKeR1* was significantly different from WT and complemented strains (*p* < 0.01).

## Data Availability

The original contributions presented in the study are included in the article/Appendix A, further inquiries can be directed to the corresponding author.

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
