# Peer review of "The Kelch Repeat Protein VdKeR1 Is Essential for Development, Ergosterol Metabolism, and Virulence in Verticillium dahliae"

_jof, 2024, doi:10.3390/jof10090643_

Round 1

Reviewer 1 Report

This work expands our knowledge of the role of Kelch proteins in fungi, in general, and in pathogenic fungi, in particular.  It provides information about the potential role that VdKeR1 plays in ergosterol metabolism.   The work also supports a role for this protein in pathogenicity, a role shared with homologues in other fungi.  Another important finding is its role in melanin production and microsclerotia formation, roles which to my knowledge, had not been described for other fungal Kelch homologues.

See attachment

Reviewer 2 Report

The subject of your manuscript is interesting and addition in scientific field. 

1. Keywords:

Change all keywords to suitable words

2.  Line 82-85:”This study” to end:

Rewrite as what are you aimed of this study

3. Line102: “[Add the Ref].”

Check it

4.Why did not the authors write conclusion?

Write the conclusions in separation section

Round 2

Reviewer 1 Report

Relevance is as stated in my previous review of this manuscript.  It has not changed.

See attached file

Reviewer 2 Report

The subject of your manuscript is interesting and addition in scientific field. 

1. Keywords:

Change all keywords to suitable words

2.  Line 99:”This study investigated the biological function of”:

Rewrite as what are you aimed of this study

3.Why did not the authors write conclusion?

Write the conclusions in separation section, without repeating the results

Round 3

Reviewer 2 Report

NA

1. Change two Keywords:

Fungal development and pathogenicity